# Milking Temperament and Its Association with Test-Day Milk Yield in Bulgarian Murrah Buffaloes

**DOI:** 10.3390/ani14070987

**Published:** 2024-03-22

**Authors:** Tihomira Stepancheva, Ivaylo Marinov, Zhivka Gergovska

**Affiliations:** Department of Animal Husbandry—Ruminant Animals and Animal Products Technologies, Faculty of Agriculture, Trakia University, 6000 Stara Zagora, Bulgaria; tihomira_78@abv.bg (T.S.); gergovskaz@abv.bg (Z.G.)

**Keywords:** Bulgarian Murrah buffaloes, milking temperament, test-day milk yield, lactation curve

## Abstract

**Simple Summary:**

The increase in the productivity of dairy buffaloes is one of the main factors for the introduction of intensive management systems and the mechanization of daily production activities. Machine milking can be considered as a critical point, as a process that is repeated at least twice a day and includes physical and psychological stressors: machine settings and maintenance, a new environment, calf separation and closeness to a person are potential sources of chronic stress, which negatively affect some aspects of buffalo behavior, welfare and productivity. Buffalo behavior during milking is an extremely important tool for determining the temperament of the animal, which in turn is essential for determining the level of welfare and conditioning of this activity, thus minimizing the problems and consequences for both animals and people who serve them. There are no recent studies to evaluate the milk temperament score of Bulgarian Murrah buffaloes in Bulgaria and how it affects their milk productivity, which is why we conducted our study.

**Abstract:**

The goal of this research was to evaluate milking temperament and its relationship with test-day milk (TDMY0) yield in Bulgarian Murrah buffaloes. This study involved 90 buffalo cows reared under a tie-stall production system which were milked twice a day with a milking pipeline. The behavioral responses of the buffaloes were reported during preparation for milking and during actual milking. The average temperament score during preparation for milking was 1.83, and 1.93 during milking itself. The most common reaction was leg lifting (18.9%), followed by cows moving on the stall bed (10%), definite kicking (9.9%), and 13.3% managing to remove the milking cluster during milking. The frequency of buffaloes showing adverse reactions (scores 4 and 5) increased considerably during milking compared to preparation for milking. Repeated scoring of temperament during the same lactation did not show a significant difference in the frequency of temperament assessments both in preparation for milking and during milking. The minimal difference may be due to the accuracy of the assessment or a momentary change in the condition of the animals during the two scorings. Cows with the most unwanted milking behavior (scores 5 and 4) had the highest LS means for TDMY, 8.18 kg and 7.65 kg, respectively. The reasons for these buffaloes remaining until later lactations was their high milk yield and the injection of oxytocin before milking, which helps them to be fully milked.

## 1. Introduction

In recent decades, due to increased economic interest, buffalo farming in a number of countries has shifted from traditional extensive productive systems to intensive systems developed for dairy cattle [1]. One of the main factors for the introduction of intensive management systems and the mechanization of daily production activities is the increase in the productivity of dairy buffaloes [2]. Most of the management techniques introduced are based on the experience gained from dairy cattle, including machine milking, artificial insemination, yards and free-stall farming systems. These practices, which expose animals to environmental change, lead to physical and physiological stress and negatively affect some aspects of buffalo behavior and welfare [1,3]. Knowing the behavior of buffaloes is important for farmers to reduce the impact of stressors on welfare and productivity. Machine milking can be considered as a critical point, as a process that is repeated at least twice a day and includes physical and psychological stressors: machine settings and maintenance, a new environment, calf separation and closeness to a person are potential sources of chronic stress [4,5,6].

Rangel et al. [7] show that monitoring the behavior of buffaloes during milking is important because this indicator can be used to assess the welfare and conditioning of these animals during milking, including assessing the characteristics of their temperaments. Behavior during milking is an extremely important tool for determining the temperament of the animal, which in turn is essential for determining the level of welfare and conditioning of this activity, thus minimizing the problems and consequences for both animals, as well as for the people who serve them [8].

Although lactating buffaloes are known to be more sensitive to manipulation during milking than dairy cows, there are few studies focusing on the reactivity to manipulations, as most studies are mainly focused on the physiology of stress in buffaloes and milk let down [2,9]. Regarding the relationship between the milking temperament and milk yield of buffaloes, there are results from the studies of different authors and under different breeds and conditions. However, there is no consensus on the relationship between the milking temperament of buffaloes and their productivity. A number of authors found a positive correlation between milking temperament and productivity, showing that calm animals have higher productivity. Patel et al. [10], indicated a negative correlation between temperament score and milk yield. Carvalhal et al. [11] hypothesize that the most reactive buffaloes are more susceptible to stress and therefore less efficient in redirecting their energy towards milk production. This also explains the lower milk yield and fat content in reactive animals. According to Patel et al. [10], temperament has a great influence on the entire milking process, and calm buffaloes perform with the best milking performance. In addition to being more suitable for milking, such buffaloes are also safer to service.

However, there are also studies pointing to inverse dependencies. For example, Gonzaga and Lorenzo [12] found that milking temperament did not correlate with milk yield and milking time in Bulgarian Murrah buffaloes in Brazil. Contrary to the above-cited studies, Rangel et al. [7] found that the most reactive buffaloes, i.e., those with higher scores, had a higher average daily milk yield. These conflicting results may be due to various reasons, such as different heredity of the studied groups and breeds of buffaloes, different milking conditions (presence of the calf, with or without oxytocin application), and also the combination of behavioral responses in the evaluation system used. As Prasad and Jaya Laxmi [13] pointed out, animal temperament is generally considered to be hereditary and is a result of the physical and nervous organization of the animal, but it is also influenced by external factors, [14]. Because of the non-unidirectional relationships between milking temperament and productivity, selection for milk yield alone cannot be expected to eliminate animals with undesirable milking temperament.

A study on milking behavior (temperament) in Bulgarian Murrah buffaloes has not been conducted in Bulgaria. Considering the interest shown in this breed by other countries from around the world, such a study would be of interest.

The aim of this study was to assess milking temperament and to find its relationship with productive traits in the cows of buffalo breed Bulgarian Murrah.

## 2. Materials and Methods

### 2.1. Study Site and Study Animals

This study was conducted in a buffalo farm in the village of Dimitrievo, Bulgaria, where 120 dairy buffaloes of the Bulgarian Murrah breed were housed. Cows were housed under the conditions of a tie-stall housing system; and during the appropriate seasons, the animals grazed on pasture. After calving, the calves were separated from the cows and not allowed to suckle. Dry cows were housed in a separate group, using the same housing system. Cows were milked twice a day on site with a milking pipeline. At each milking, the cows were injected with oxytocin for faster milk let down and more complete milking. Before milking the buffaloes, udder preparation was carried out, including washing with a disinfectant, drying and checking for clinical mastitis.

This study included 90 lactating buffaloes of the Bulgarian Murrah breed. The milking temperament score was assessed twice during the year—in July and in November. All buffaloes from 30 to 240 days in milk (DIM) were scored. In order to test the assumption that as lactation progresses, a change in animal behavior can be observed, a re-evaluation of temperament (in November) was performed on 41 buffaloes evaluated in July and still lactating in November.

The data for milk yield TD records and for origin of all animals were taken from the farm documents. The number of milk yield TD records of the cows included in this study was 682.

Cows are presented in groups depending on their lactation number, respectively, from 1st to 4th—13 cows; from 5th to 7th—41 cows; from 8th to 10th—26 cows; and 11th and more lactations—10 buffalo cows. Due to the small number of first- and second-lactation buffaloes (5 in total), animals up to the 4th lactation are represented in one group.

For the purpose of this study, the calvings were grouped by season of calving, respectively: winter—from December to February; spring—from March to May; summer—from June to August; and autumn—from September to November.

### 2.2. Temperament Scoring Methods

Temperament scoring was performed according to the methodology of Tulloh [15] and Dogra et al. [16]. The temperament scoring was performed in two parts—during the attaching the milking cluster (direct contact with a person) and during milking, on a scale from 1 to 5, as shown in Table 1.

To differentiate between kicking and lifting the rear legs, we have adopted the definitions given by Rushen et al. [17], Hemsworth et al. [18], Van Reenen et al. [4] and De Rosa et al. [3]:

Leg steps—whenever one hoof was lifted less than 15 cm off the ground.

Kicking—the hoof is raised at least 15 cm off the ground towards the level of the udder or towards the milkman.

Based on the reported reactions, the final scores for temperament were formed (during preparation for milking and during milking itself).

### 2.3. Statistical Analysis

For the statistical processing of the data, the corresponding modules of the statistical packages of STATISTICA 6 and EXCEL were used.

The following model was used to assess the influence of milking temperament on test-day milk yield:
Y_ijklm_ = μ + S_i_ + L_j_ + TS_k_ + K_l_ + e_ijklm_(1)
where Y_ijklm_ is the dependent variable (test-day milk yield); μ is the model mean; S_i_ is the effect of calving season; L_j_ is the effect of lactation number; TS_k_ is the effect of temperament score at milking; K_l_ is the effect of test-day number; and e_ijklm_ is the effect of the non-included random factors.

By means of the analysis of variance (ANOVA) for the model, the least squares means (LSM) were obtained by classes of the fixed factors.

## 3. Results

### 3.1. Average Values for the Productive Traits Performance

Table 2 presents the average values for the main productive traits of the buffaloes included in this study. The average milk yield for standard lactation was 2245.37 kg, with an average percentage of milk fat of 7.77% and 4.34% of milk protein. The variation in the average milk yield was over 1000 kg, the minimum reported yield was 1805.85 kg, and the maximum was 2947.43 kg. This was due not only to the difference in the heredity of the animals, but also to the different age and duration of lactation (from 240 to 305 days). In the milk composition traits, the variation was smaller—in the fat percentage, it was from 7.46% to 8.24%; and in the protein percentage, it was even smaller—from 4.26% to 4.58%. The average milk yield for a test day was 7.34 kg, with a variation from 3.92 to 11.66 kg. The reported milk yield test-day records were within 10 for lactation.

### 3.2. Temperament Score during Milking Cluster Attaching and during Milking

A temperament score that included two parts—score during milking cluster attaching and score during actual milking—was recorded for the lactating buffaloes at the farm. The average values and variation in the scores are presented in Table 2. The average temperament score during cluster attaching was 1.83, which was a very good score as the average for the herd. The average temperament score during milking, 1.91, was slightly higher as the difference was small and not statistically significant.

Figure 1 presents the individual behavioral reactions of the cows reported during milking, on the basis of which the temperament score during milking was formed. The highest was the percentage of cows that reacted by lifting their leg (18.9%), followed by animals that move on the bed during milking (10%) and those that definitely kick (9.9%). A relatively high percentage of cows managed to remove the milking cluster during milking (13.3%). A small percentage of animals reacted to touch (4.4%) and stepped from foot to foot (5.4%). A total of 16.7% of cows defecated and urinated during milking. Some of the animals showed one certain reaction and others showed several of these behavioral reactions.

Similar behavioral reactions have been reported during attaching of the cluster, but due to the shorter time, some of the reactions indicated during milking cannot be manifested, such as stepping and removing the cluster. The main behavioral reactions reported during attaching the milking cluster and on the basis of which this score was formed were: moving on the bed, lifting a leg and kicking. Figure 2 shows the difference in the percentage of animals, with the three main behavioral reactions during attaching the milking cluster and during milking itself. A higher percentage of cows manifested leg lifting and kicking during the attaching of the milking cluster, compared to milking itself, 27.8% and 13.3%, respectively, during attaching and 18.9% and 9.9% during milking. Almost equal percentages of cows have tried to move on the bed during cluster attaching and during milking, 3.3% and 4.4%, respectively.

The reported difference in these behavioral responses indicates that the cows showed slightly more anxiety and unwanted reactions during contact with the milker (milking cluster attaching) than during milking itself. In most cows, the unwanted reactions were repeated, with some cows manifesting them with varying intensity (frequency) during both operations.

On the basis of the reported behavioral reactions and data from the milkers for the specific behavior of the individual animals, the scores for temperament during milking preparation and milking were rated.

Table 3 shows the frequency of milking temperament scores in buffalo cows during preparation for milking and milking. During preparation for milking, the animals with a score of 1, calm (65.56%) were the most frequent, followed by those with a score of 2, slightly restless (21.11%). During milking, a slightly lower frequency of animals with temperament scores of 1 and 2 was reported at 55.56% and 17.78%, respectively, with an increasing frequency of those with scores of 3 to 5. During milking, the frequency of cows with adverse reactions (grades 4 and 5) considerably increased, compared during preparation. The frequency of animals attempting or succeeding in removing the milking cluster during milking was increased substantially (scores 4 and 5).

The correlation between the two scores, during cluster attaching and during milking, was positive, with an average value of r_p_ = 0.42 and significant (*p* < 0.05). This indicates that most animals exhibiting undesirable behavior during cluster attaching have a similar behavior during milking too. Also, during milking, unwanted reactions associated with kicking to the milking cluster and its removal (scores 4 and 5) occur with greater frequency, which cannot be reported during preparation.

A total of 41 of the cows included in this study were scored for temperament during milking cluster attaching and during milking twice within lactation. The first scoring was at the beginning of lactation until approximately 150 DIM, and the second towards lactation end. The frequency of temperament scores both in preparation for milking and during milking was almost the same for both scorings, as shown in Table 4. In the second scoring during preparation, a slightly higher frequency of animals with a score of 3 and a lower frequency of those with scores of 4 and 5 were reported compared to the first temperament score. Milking temperament scores showed even less change in the frequency. In the second scoring, the frequency of animals with scores of 2 and 5 decreased slightly and the frequency of those with a score of 3 increased. No statistically significant differences were found between the scores at the two scorings in both indicators. The difference in the two consecutive scores in some of the animals was minimal, which may be due to the accuracy of the scoring, or a momentary change in the condition of the animals during the scorings. The obtained result indicated that the temperament score according to the applied system allows for correct and relatively constant animal scores, showing a very small change depending on the period of scoring.

### 3.3. Factors Affecting the Temperament Score during Cluster Attaching and during Milking

There were two permanent milkers on the farm. Cows enter the designated spot for them at the milking sites after calving, thus they may be milked by different milkers periodically during different lactations. The difference in the average temperament scores depending on the milker was minimal—0.1. This showed that the two milkers had the same behavior and attitude towards the animals; and given the small number of cows with unwanted temperament scores, it can be said that they treat the animals well.

In conversation with the milkers, this fact was confirmed—cows showed consistency in temperament during milking (behavioral reactions); from one lactation to the next, the changes were very weak, and can be in both directions.

Table 5 shows the frequency of distribution of buffaloes with different milking temperament scores depending on the parity. There were very few young animals in first and second lactation at the time of evaluation; therefore, we present them in one group from first to fourth lactation. In all groups, the frequency of animals with scores of 1 and 2 was the highest. The highest frequency of buffaloes with undesirable scores of 4 and 5 was reported in the group from 5th to 7th parity, followed by the oldest group, above the 11th parity. In the groups up to 4th and from 8th to 10th parity, animals with a score of 5 were not reported, and the frequency of those with a score of 4 was low. From the presented frequency of distribution of buffaloes with different scores by parity groups, it can be seen that there was no definite dependence between the age of the cows and their temperament during milking. In conversation with the milkers and the owner of the farm, it was found that buffaloes with bad habits during milking had good milk yield and regardless of the problems they created during milking, they were not culled.

### 3.4. Effect of Milking Temperament on the Shape of the Lactation Curve

Based on the data from the monthly TDMY records, the effect of the controlled factors (calving season, number of lactations, number of TDMY records and milking temperament) on the TDMY was studied, as shown in Table 6.

All factors included in the model had a significant effect (*p* < 0.001) on test-day milk yield, including the milking temperament. On the trait test-day fat % in milk, the calving season (at *p* < 0.01), number of lactations (at *p* < 0.05) and number of test-day records (at *p* < 0.001) had a significant effect, but the temperament score did not. In the trait test-day protein % in milk, none of the factors included in the model had a significant effect.

The significant effect of calving season, number of lactations and successive milk yield test-day records has been widely studied and proven. The inclusion of these factors in the statistical model was to obtain the corrected effect (LS mean) of milking temperament score on milk yield and milk composition for a Test day. Table 7 presents the LS means from the test-day milk yield model depending on the temperament score. The highest test-day milk yield was from cows with a temperament score of 5–8.18 kg, which was almost 1 kg of milk more than the mean milk yield of cows with scores from 1 to 3 (from 7.21 to 7.37 kg). Next in test-day milk yield were cows with an unwanted milking temperament—score 4, with 7.65 kg test-day milk yield.

This fact largely explains the presence of cows with a greater number of lactations and with unwanted milking temperament—they stay in the herd due to good milk performance, regardless of the troubles during milking.

Based on the estimated LS means for test-day milk yield by number of test-day records for the group of cows with different temperament scores, the lactation curves are presented on Figure 3. Generally, all groups showed a slower peak of lactation—the 3rd to 4th month after calving. This is typical of buffaloes, unlike cattle.

The figure clearly shows the difference in the shape of the lactation curve of buffaloes with temperament scores from 1 to 3 and of those with scores of 4 and 5. In cows with scores of 1, 2 and 3, the lactation curves were similar both in shape and in maximum and minimum test-day milk yield. In all these animals, the maximum milk yield was 7.5 to 8.0 kg and was maintained for several months with small variations. A more serious decrease in milk yield was reported after the 6th lactation month. The level of test-day milk yield at the beginning of lactation was the same in all three groups—approximately 7 kg and it sensibly decreased by the 8th lactation month—to 6.50 kg.

The lactation curve in buffaloes with a temperament score of 5 had the smoothest shape. The maximum milk yield reached by them was the highest—over 8.5 kg. The decrease in milk yield was also at much lower rates than in buffaloes with other temperament scores, maintaining a level of a little over 8 kg by the 8th month. The lactation curve in buffaloes with a temperament score of 4 was with the most undesirable shape—steep, reaching the peak and a sharp decrease in milk yield after the peak. Although they reached almost the same milk yield as cows with a score of 5 at the peak, their milk yield after that dropped to the lowest level—below 6.5 kg.

## 4. Discussion

The reported productive performance of the buffaloes included in this study (Table 2) was around the average for the breed in Bulgaria. According to EASRA [19], the average milk yield of the controlled buffaloes of the Bulgarian Murrah breed in Bulgaria was 1500–2500 kg, with 7–9% fat and 4.6% protein in the milk. Dhillod et al. [20] also showed a relatively high average milk yield for 305-day lactation in Indian Murrah buffaloes in the state of Haryana—2604.38 kg. According to Matera et al. [21] in Italy, the milk yield of Italian Mediterranean buffalo for 2022 reached 2350 kg for lactation, with a fat% of 7.72 and a protein% of 4.65.

Regarding the milking temperament, other authors have found results similar to ours in other buffalo breeds. Gonzaga and Lorenzo [12] found an average temperament score of 1.68 in a study of 60 lactating Bulgarian Murrah buffaloes in Brazil. The authors indicated that no significant difference was found between the average temperament scores for morning and evening milking, 1.66 for morning milking and 1.70 for evening milking. Patel et al. [10] showed an average milking temperament score of 1.89 in buffaloes of the breed Mehsana.

In relation to the results presented in Figure 1, the elimination frequency (urination) is more closely related to buffalo temperament and is considered an indirect measure of fear [22]. On the other hand, Saltalamacchia et al. [23] found that the frequency of oxytocin injecting correlates with stepping from foot to foot and kicking, and is highly repeatable when tested again after five months.

It should be noted that all buffaloes were constantly injected with oxytocin before milking, which affects not only the time for milk letdown and milking completeness, but also the behavior of buffaloes. In a study of 250 Italian buffalo cows, Cavallina et al. [24] reported significant differences in the percentage of different behavioral responses during milking depending on whether or not oxytocin was injected. When oxytocin was injected, 31.7% of cows kicked, 34.7% removed the cluster, 23.1% stepped from foot to foot, 100% defecated and 36.8% urinated during milking. Without oxytocin injection these reactions were much more common: kicking—68.3%, removal of the cluster—65.4%, stepping from foot to foot—76.9% and urination—63.2%.

Similar results were reported by Patel et al. [10] in an experiment with 24 Mehsana buffalo cows, where the highest percentage of cows (55.56%) had a calm temperament, followed by nervous (26.74%), aggressive (9.25%) and restless cows (8.45%). The results of a study by Thomas [25] in Murrah buffaloes showed that almost 50% of cows were calm during milking. Approximately 7% of the animals were aggressive and the rest were classified as restless or nervous cows. The results of a study by Choudhary et al. [26] in 102 Murrah buffaloes (21 primiparous and 81 older) showed that 36.27% were calm, 27.45% were slightly restless, and 29.41% showed restless behavior. Only 6.86% were aggressive, but no nervous cows (score 5) were reported.

There are conflicting data regarding the relationship between parity and milking temperament score (Table 5). Some authors found a statistically significant effect of parity or age in dairy cows on temperament score. Uetake et al. [27] found a moderate relationship between cow age and milking adverse reactions (rp = −0.28, r < 0.01). According to Mounaix et al. [28], age is an important factor in temperament, as it is related to the greater experience of animals. This is especially true of cattle: older animals show a much greater tolerance for unpleasant manipulations because they have experienced them before. On the other hand, Sewalem et al. [29] found that very nervous Holstein cows were 26% more likely to be culled than very calm cows. This in turn also results in a lower percentage of animals with undesirable milking temperaments present in later lactations. Sriranga et al. [30] reported a significant difference in temperament score between primiparous (2.12) and multiparous (1.27) buffaloes of the Surti breed. Dash et al. [31] found that the temperament score was highest (1.96) in fourth-lactation buffaloes, with a mean temperament score of 1.78 for all buffaloes studied. This result is similar to ours; and in our opinion, the reason was that there were different animals in the different groups by parity. The reduction in stress responses from one lactation to the next can in all probability be reported if the same animals are followed over time. It is also likely that there were young buffaloes with undesirable milking temperament but combined with low milk yield that were culled so that only those with high milk yield and temperament scores of 4 and 5 remained in the herd, but they were in different lactations, without relation to their sequence.

Regarding the dependency of temperament score on DIM, Rangel et al. [7] did not find a decrease with advancing lactation due to the adaptation of buffaloes to the milking environment. The authors report that buffaloes had good milk yield, even when they had high milking temperament scores, believing that this can be explained by the good milking practices applied in the studied farms.

The reactions we reported in scores of 2 and 3 were mainly stepping from foot to foot, moving on the bed, twitching and backing away during servicing. These reactions are defined by a number of authors as reactions of fear rather than aggression, such as lifting a leg, kicking and attempts to remove the milking cluster, to which scores of 4 and 5 correspond. Stepping foot to foot or “dancing” of cows during milking was associated with fear, increased heart rate and increased concentration of cortisol in milk, while kicking during preparation and milking was not initiated by nervousness or restlessness and was not a manifestation of cowardly cows, but rather of aggressive and disobedient ones [12,28].

Regarding the relationship between temperament during milking and the milk yield of buffaloes (Table 7), there have been quite contradictory results from studies by different authors and under different breeds and conditions. For example, Gonzaga and Lorenzo [12] found that the temperament during milking does not depend on the milk yield of the animals and on the time for milking in Bulgarian Murrah buffaloes in Brazil. The study by Mallick et al. [32] showed that the milking temperament of Bulgarian Murrah buffaloes was not influenced by the level of milk yield and milking time.

In contrast, according to Prasad and Jaya Laxmi [14], calm and slightly restless buffaloes have a significantly higher average daily milk yield than other categories. The authors conclude that the declining amount of milk from calm to aggressive and nervous animals may be due to the fact that under optimal milking conditions, the calm cows do not retain any milk, while other categories have retained milk due to adrenaline secretion. This may be the reason for the negative correlation between the temperament and average daily milk yield according to Patel et al. [33].

Rangel et al. [7] reported results similar to ours. In assessing behavioral responses individually and in relation to productive traits, they found that the most reactive buffaloes, i.e., those with higher scores, had a higher average daily milk yield. Sullivan and Burnside [34] found that milking temperament had a negative correlation with milk productivity (−0.17), but had a high correlation with feeding aggression, which in turn was positively correlated to milk yield (0.23). In summary, this means that cows showing more nervous reactions during milking (scores 4 and 5) also had a more aggressive feeding behavior compared to the other animals in the group. This provides them with longer time for and greater food intake and therefore has a positive effect on productive performance. Maffei et al. [35] found that cows with a more aggressive temperament were more reactive, mobile, and had a higher hierarchical rank in the groups.

In addition to the above-mentioned authors, the reported dependency between temperament and milk yield from our study may be due to the fact that all cows were injected with oxytocin before milking, which reduced the probability of milk retention due to problematic milk letdown and incomplete milking in animals that were more reactive. Saltalamacchia et al. [23], when monitoring the behavior of buffalo cows during milking and injection of oxytocin in the whole herd of all animals during lactation, only found difficulties in the milk letdown in primiparous. Oxytocin injection can be used as a target indicator of well-being in buffalo Cavallina et al. [25].

The reported relationship between daily milk yield and temperament score (Table 7) showed that the reason cows with a highly reactive temperament (scores 4 and 5) remain in the herd until very late lactations was definitely their higher milk yield. The reason for their complete milking was the injection of oxytocin before milking. A number of authors point out that one of the reasons why the more reactive animals have lower milk yield is that milk retention and incomplete milking were reported for them. Patel et al. [33] found that calm buffaloes had the highest milk yield and higher speed of milking. The aggressive animals were milked longer and had the lowest milk yield, followed by the nervous and restless buffaloes. The decreasing milk yield and milking speed from calm towards aggressive animals may be due to the fact that within the optimal milking time, calm animals do not retain any milk, and the other categories retain it in increasing order.

Rangel et al. [7] also indicated that it was expected that the advancing of lactation will lead to a decrease in temperament score due to the adaptation of buffaloes to the milking environment. However, this has not happened. Buffaloes with high temperament scores remained unchanged and had good milk yield.

These conflicting results may be due to different reasons, such as different heredity of the studied groups and breeds of buffalo, milking conditions (presence of the calf, with and without oxytocin injection), as well as the combination of the various behavioral responses in the scoring system used.

According to Patel et al. [10], temperament has a great influence on the whole process of milking. Therefore, calm animals are desirable for better milking, for faster and complete removal of milk from the udder and for a shorter milking duration. In addition to being more suitable for milking, such buffaloes are also safer to service.

## 5. Conclusions

The average values for the productive traits of the buffaloes included in this study were very good, within the average for the productive indicators of the breed Bulgarian Murrah in Bulgaria—milk yield 2245.37 kg, 7.77% fat and 4.34% protein in milk for 305 day lactation, respectively. The average temperament scores during preparation for milking and milking itself were similar and with minimal difference. The frequency of buffaloes showing adverse reactions (scores 4 and 5) increased considerably during milking compared to preparation for milking. In most buffaloes, adverse reactions recurred, some with different intensities (frequency) for the two operations. This shows that most animals that exhibit undesirable behavior during preparation for milking also exhibit similar behavior during milking. During milking, adverse reactions associated with kicking the milking cluster and removing it (scores 4 and 5) occurred with greater frequency than during preparation for milking.

The difference between two consecutive temperament scores during the same lactation was minimal, which may be due to the accuracy of the assessment or the momentary change in the condition of the animals during the assessment. This insignificant difference in the two consecutive temperament scores shows that no change in the behavior of the animals was observed as lactation progresses, neither in a positive (habituation) nor in a negative direction. The obtained result shows that the evaluation of the temperament according to the applied system allows correct and relatively constant scores of the animals, showing very little change depending on the time of scoring. This allows temperament scoring to be performed once during lactation for selection purposes. From the frequency distribution of buffaloes with different temperament scores and groups by parity, no definite relationship was reported between the age of the buffaloes and their temperament at milking. This may be due to the fact that these were different animals at different parities. In order to report a possible change in milking temperament in buffaloes with successive parities, it is necessary to follow the same animals for several successive calvings.

Cows with the most unwanted milking behavior (scores 5 and 4) had the highest LS means for TDMY, 8.18 kg and 7.65 kg, respectively. The buffaloes with temperament scores from 1 to 3 had milk yields that varied little and that were lower than for among animals with scores of 4 and 5—from 7.21 to 7.37 kg, respectively. Their high milk yield is also the reason why, despite the problems with their service, they remain in the herd until later lactations. Another factor was that before milking, all animals were injected with oxytocin, which was a prerequisite for buffaloes with an undesirable temperament to have no problems with milk retention, incomplete milking and subsequent problems.

## Figures and Tables

**Figure 1 animals-14-00987-f001:**
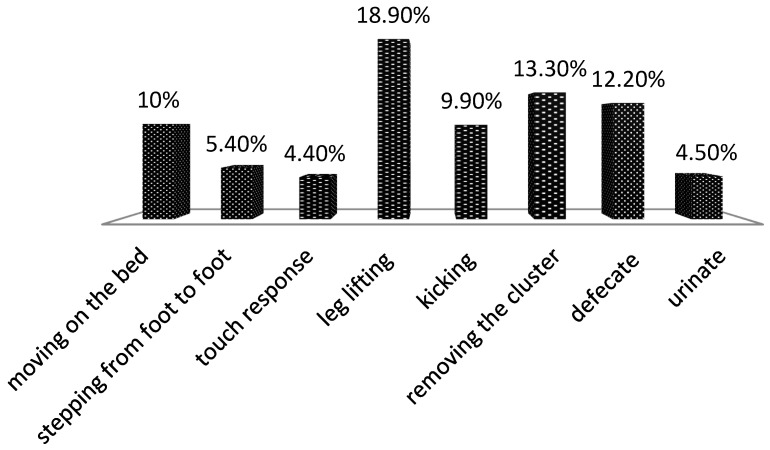
Reported behavioral reactions in buffalo cows (*n* = 90) during milking (in %).

**Figure 2 animals-14-00987-f002:**
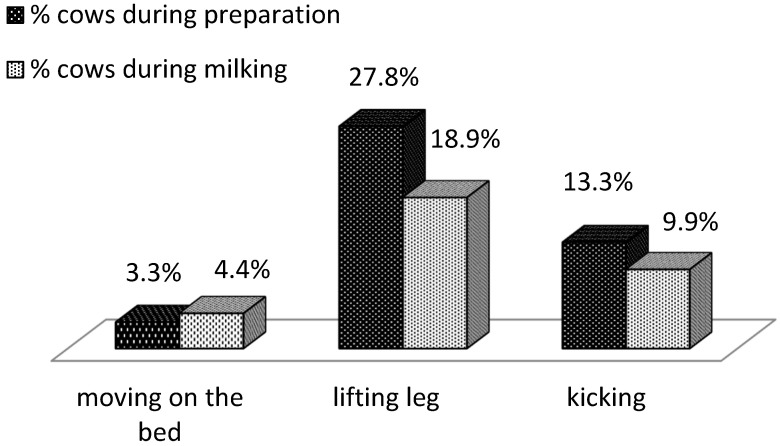
Difference in the percentage of behavioral reactions during milking preparation and during milking (*n* = 90).

**Figure 3 animals-14-00987-f003:**
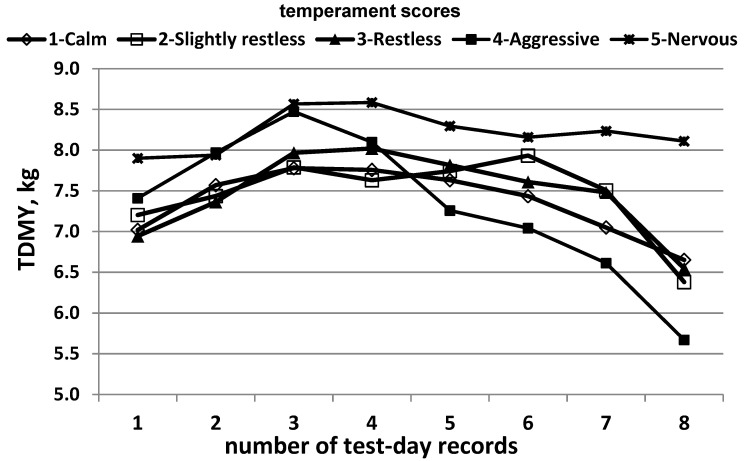
Lactation curves depending on temperament score.

**Table 1 animals-14-00987-t001:** Definition and temperament scores during buffalo cows milking.

Description	Definition	Score
Very calm, never causes any problems, extremely docile during milking and preparation, “ideal” for milking	Calm	1
Stand still, not bothered by preparation or milking, but can move often, shifting weight from one leg to the other, can flicks tail occasionally, causes very few problems	Slightly restless	2
Generally calm, but moves on the bed, moves often, can sometimes lift leg during preparation or milking, but does not kick, often flicks tail or seem restless from time to time	Restless	3
Looks very restless during preparation or milking, kicks towards the milkman, often steps from one side to the other, trembles when a hand is placed on it	Aggressive	4
Shows great anxiety during preparation or milking, kicks the milkman and the cluster, fights with violence, sometimes trembles when a hand is placed on the back	Nervous	5

**Table 2 animals-14-00987-t002:** Average values for the main productive traits of the buffaloes included in this study.

Trait	Average Values and Variation, *n* = 90			
	x ± SE	SD	Min	Max
305 days milk yield, kg	2245.37 ± 27.60	261.86	1805.85	2947.43
test-day milk yield, kg	7.34 ± 0.04	1.16	3.92	11.66
fat %	7.77 ± 0.02	0.16	7.46	8.24
protein %	4.34 ± 0.01	0.05	4.26	4.58
milking temperament score	1.91 ± 0.14	1.29	1	5
temperament score during milking cluster attaching	1.83 ± 0.12	1.12	1	5

**Table 3 animals-14-00987-t003:** Frequency of milking temperament scores (%) during preparation and milking (*n* = 90).

Temperament Score	During Preparation	During Milking
1	65.56	55.56
2	21.11	17.78
3	8.89	12.22
4	3.33	6.67
5	1.11	7.78

Scores definition: 1—calm, 2—slightly restless, 3—restless, 4—aggressive, and 5—nervous.

**Table 4 animals-14-00987-t004:** Frequency of occurrence (%) for temperament scores during preparation and during milking at the first and second scoring during the lactation (*n* = 41).

Temperament Score	During Preparation		During Milking	
	First Scoring	Second Scoring	First Scoring	Second Scoring
1	36.59	36.59	36.59	36.59
2	4.88	4.88	3.66	2.44
3	2.44	7.32	4.88	7.32
4	4.88	1.22	1.22	1.22
5	1.22	0	3.66	2.44

Scores definition: 1—calm, 2—slightly restless, 3—restless, 4—aggressive, and 5—nervous.

**Table 5 animals-14-00987-t005:** Frequency of milking temperament scores in buffalo cows at different parities (*n* = 90).

Milking Temperament Score	Number of Parities			
	Up to 4th	From 5th to 7th	From 8th to 10th	Above 11th
1	19.51	51.21	41.46	9.75
2	2.44	19.51	9.76	7.31
3	7.32	9.76	9.76	0
4	2.44	7.32	2.44	2.44
5	0	12.20	0	4.88

Scores definition: 1—calm, 2—slightly restless, 3—restless, 4—aggressive, and 5—nervous.

**Table 6 animals-14-00987-t006:** Analysis of variance for the influence of the controlled factors on the test-day productive traits.

Sources of Variation	Degrees of Freedom	Test-Day Milk Yield, kg		Test-Day Fat %		Test-Day Protein %	
	(n-1)	MS	F (*p*)	MS	F (*p*)	MS	F (*p*)
Total for the model	28	14.48	18.76 (***)	0.34	3.09 (***)	0.04	1.3 (-)
Calving season	3	4.86	6.30 (***)	0.58	5.3 (**)	0.02	0.8 (-)
Number of lactations	12	7.96	10.32 (***)	0.22	2.0 (*)	0.04	1.4 (-)
Number of test-day records	9	22.43	29.05 (***)	0.58	5.3 (***)	0.05	1.8 (-)
Milking temperament score	4	5.31	6.88 (***)	0.16	1.4 (-)	0.01	0.5 (-)
Error	653	0.77		0.11		0.11	

* significance at *p* < 0.05; ** significance at *p* < 0.01; *** significance at *p* < 0.001; - no significance.

**Table 7 animals-14-00987-t007:** LS means of test-day milk yield depending on milking temperament score.

Milking Temperament Score	Number of Test-Day Records	Test-Day Milk Yield, kg
	*n*	LSM ± Se
1	405	7.21 ± 0.057
2	108	7.35 ± 0.113
3	82	7.37 ± 0.099
4	36	7.65 ± 0.182
5	51	8.18 ± 0.164

Scores definition: 1—calm, 2—slightly restless, 3—restless, 4—aggressive, and 5—nervous.

## Data Availability

All data presented in this study are available on request from the corresponding author. The data are not publicly available due to the fact that primary data obtained during animal observation were recorded in a paper checklist.

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
