# Peer review of "Milking Temperament and Its Association with Test-Day Milk Yield in Bulgarian Murrah Buffaloes"

_animals, 2024, doi:10.3390/ani14070987_

Round 1
Reviewer 1 Report
Comments and Suggestions for Authors
1) Brief summary
Exports of buffalo milk to the USA, New Zealand, Germany and France are steadily increasing. Studies into the factors likely to influence this production are therefore increasingly necessary. This very interesting publication focuses on an important aspect, namely the animal's behavior before and during milking, and its effects on the quantitative and qualitative aspects of milk production. As many clinicians, I am unfamiliar with buffalo milk production. Reading the publication, I have learned many new things.
2) General concept comments.
The study is very well introduced and described. All the different results are very well presented and also discussed through a very large bibliography. Use of oxytocin seems to be quite necessary to decrease the stress effect associated with milking. The presence of a control group have provided additional information. Moreover, It would have been interesting to measure the milking time.
3) Specific comments
37 Attending some people are unfamiliar with the buffalo milk production, it could be interesting to give some general informations on this production and lenght of lactation
67 Bubalus bubalus ? Which difference with the breed used in Italy (Bufala meditteranea italiana)
71 What kind of udder preparation is used (expulsion of foremilk, washing and rying of the teats) …?
72 which dose of oxytocin is given ?
135 1.91 and not - 1.91
146 have you observed some negative effect of kicking for the milkers ?
157 how long is the milking time on average ?
251 Do you think that higher is the milk production, longer is the milking time and higher is the probability of unwanted behavior ?
407 Do you think that behaviour could be a parameter for genetic selection ?
Reviewer 2 Report
Comments and Suggestions for Authors
General comment
The study by Stepancheva et al. explored the relationship between buffalo temperament and test day milk yield. The study is particularly significant given the increasing importance of buffaloes in global dairy production and the trend towards farming intensification. While the manuscript contains important information, it requires substantial revisions in several key sections, including the introduction, materials and methods, results, and discussion. Please see the following comments and suggestions to improve the manuscript.
Title
Please consider shortening the title to "Milking Temperament and its Association with Test Day Milk Yield in Bulgarian Buffaloes.”
Abstract
The abstract lacks a concluding sentence to summarize the key findings and implications of the study. Consider adding a sentence or two to summarize the main outcomes of the research and its practical applications in the field.
Introduction
The introduction section provides important background information on the intensification of the buffalo farming systems, setting the stage for the study. However, it lacks sufficient explanation on why assessing temperament is necessary. It would be beneficial to expand this section by elaborating on how temperament in buffaloes is associated with productivity and the importance of assessing temperament beyond just indicating stress. Consider incorporating a brief discussion on how temperament affects factors such as milk production, reproductive performance, and overall herd management.
Materials and Methods
The material and methods section needs considerable improvement. It is advisable to divide this section into subheadings, like study site and study animals, temperament scoring methods and statistical analysis. Update the details of the animals, the study site, the actual months during which the study was conducted.
Line 75-76: The sentence is not clear. What the information is about.
Line 79-82: The classification of buffaloes into parities should be based on physiological differences. For instance, categorizing them into first parity, second parity, and greater than 2nd parity could be more informative. This is important because primiparous buffaloes may respond differently to machine milking compared to animals in other parities. Additionally, please present the distribution of buffaloes into different parities.
It appears that scoring was done twice (as mentioned in lines 183-186). It would be helpful to describe this process in the Materials and Methods section of the manuscript. Consider adding a brief explanation of why scoring was done twice and how the results were analyzed to account for this in the final analysis.
The description of temperament categories in the manuscript is unclear. To improve clarity, I recommend consulting other studies, such as reference [14], to precisely define these categories. Additionally, it would be helpful to avoid repeating the same phrases (e.g., "very calm," "absolutely calm," "very docile", all were used to define the score 1) to prevent ambiguity.
Statistical Analysis
For the descriptive statistics, presenting the percentage/frequency of temperament scores was appropriate. However, given the nature of the variable "temperament score" and its distribution, applying ANOVA on temperament scores is incorrect. I suggest using ordered logistic regression instead, as it is more appropriate for analyzing ordinal data such as temperament scores.
Results
Figure 1: Please include the total number of events (n=?) in the figure.
Figure 3: Instead of presenting the percentages, please present the frequency of temperament scores (from both the scoring time points; in early lactation and late lactation). This will provide a clearer picture of the distribution of temperament scores among the buffaloes studied.
Table 3: Instead of presenting the means and standard deviations, please present the frequency of each temperament score. This will provide a clearer representation of the distribution of temperament scores among the buffaloes studied.
Table 4 is unnecessary and may present inappropriate analysis. Consider removing it to streamline the presentation of your results and avoid confusion.
Figure 4: Consider presenting the frequency of scores with the new classification of parity of buffaloes. This will provide a clearer picture of the distribution of temperament scores among the different classifications of buffaloes based on parity.
Figure 5: It was described in the text that the scoring was done for early and late lactation (lines 183-186,), while this figure describes the data for early, mid, and late lactation. Please clarify this discrepancy and update the text accordingly.
Line 249-252: The description of the values is inconsistent and creates ambiguity. Was it 8.18 or 8,18? Please clarify the format of the values to ensure accuracy and understanding.
Instead of presenting the ANOVA table for all the adjusted variables, consider presenting the LS Means of milk yield for each temperament score with standard error. You can include the details of the adjusted variables in the table footnote for clarity.
Figure 7: The data presented in the figure does not directly associate with the temperament score. Consider either using the test day data as retrospective or prospective in relation to temperament scoring. Alternatively, clearly describe in the Materials and Methods that the buffaloes were enrolled in early lactation and were followed throughout the lactation, with the first temperament scoring done in early lactation and the last one in late lactation. As there was no such description in the Materials and Methods section, this figure does not make sense.
Discussion
The Discussion section needs some improvement.
Lines 322-344: Please clearly justify the increase in temperament score with the increase in parity. Provide a detailed explanation or refer to relevant literature to support this relationship.
Lines 362-371: Nervous animals may not necessarily be the highest-ranked animals in a herd, which would have ensured more feeding and thereby higher milk yield. Please provide another analogy or reference to support the association of higher temperament with higher milk yield.
Lines 373-376: As the manuscript describes that it could be possible that the complete removal of milk could be attributed to the oxytocin injection, which could confound the negative association of temperament with milk production in buffaloes? Consider increasing the discussion about the potential effects of oxytocin injection on milk harvesting and the potential confounding effect on the relationship of test day milk yield with temperament to address this concern.
Lines 389-394: The study by Praxedes [34] did not clearly describe the methods of temperament assessment and should be used cautiously to justify the results. Consider discussing the limitations of using this study as a reference for supporting the results.
Conclusion
The conclusion of the study was relevant but needs updating. It should concisely describe the key points of the study instead of repeating every detail. Consider summarizing the main findings and their implications for buffalo management and welfare.
Comments on the Quality of English LanguageThe manuscript requires moderate English editing to correct minor grammatical errors and enhance sentence clarity. Rephrasing certain sentences and incorporating suitable transitional phrases and words will enhance the overall flow of the manuscript. English editing AI tools could be beneficial for improving the language of the manuscript.
Round 2
Reviewer 2 Report
Comments and Suggestions for Authors
The authors have satisfactorily addressed the comments provided in the previous feedback. The manuscript can be substantially improved by enhancing the language. Professional English editing would be of great help. A couple of comments to further improve the quality of the manuscript are provided below:
The abstract needs to be reduced to about 250 words to comply with the journal guidelines.
Concisely define the temperament scores in the footnotes of tables 3, 4, 5, and 7, as well as in the caption of figure 3.
The conclusion appears to duplicate the abstract's content. It would be more effective if it concisely summarized the key findings and separated recommendations into a distinct paragraph.
Comments on the Quality of English LanguageEnglish editing is necessary to enhance the quality of the manuscript. Several sentences require rephrasing to improve clarity and simplicity. For example, the sentence in lines 24–26 could be revised as follows: "The most common reaction was leg lifting (18.9%), followed by cows moving on the stall bed (10%), definite kicking (9.9%), and 13.3% managing to remove the milking cluster during milking." Such editing of the entire manuscript would be beneficial.
Author Response
Thank you very much for taking the time to review this manuscript. Please find the responses below. The corresponding revisions in the re-submitted file are highlighted in yellow.
The Abstract and Conclusion sections have been entirely revised following the changes we made in Results and Discussion at your recommendation.
We included concise definition of the temperament scores in the footnotes of tables 3, 4, 5, and 7, as well as in the caption of figure 3.
An English language revision of the manuscript was made.
